# Inflammatory and Autoimmune Aspects of Multisystem Inflammatory Syndrome in Children (MIS-C): A Prospective Cohort Study

**DOI:** 10.3390/v16060950

**Published:** 2024-06-12

**Authors:** David A. Lawrence, Aishwarya Jadhav, Tapan K. Mondal, Kyle Carson, William T. Lee, Alexander H. Hogan, Katherine W. Herbst, Ian C. Michelow, Michael Brimacombe, Juan C. Salazar

**Affiliations:** 1Wadsworth Center, New York State Department of Health, Albany, NY 12208, USA; aishwarya.jadhav@health.ny.gov (A.J.); tapan.mondal@gmail.com (T.K.M.); kyle.carson@health.ny.gov (K.C.); william.lee@health.ny.gov (W.T.L.); 2School of Public Health, University at Albany, Rensselaer, NY 12144, USA; 3Division of Hospital Medicine, Connecticut Children’s, Hartford, CT 06106, USA; ahogan@connecticutchildrens.org; 4Department of Pediatrics, University of Connecticut School of Medicine, Farmington, CT 06030, USA; imichelow@connecticutchildrens.org (I.C.M.); mbrimacombe@connecticutchildrens.org (M.B.); jsalaza@connecticutchildrens.org (J.C.S.); 5Division of Pediatric Infectious Diseases and Immunology, Connecticut Children’s, Hartford, CT 06106, USA; kherbst@connecticutchildrens.org; 6Department of Research, Connecticut Children’s Research Institute, Hartford, CT 06106, USA

**Keywords:** inflammation, Multisystem Inflammatory Syndrome in Children, COVID, autoantibodies, cytokines

## Abstract

Multisystem Inflammatory Syndrome in Children (MIS-C) is a potentially life-threatening complication of COVID-19. The pathophysiological mechanisms leading to severe disease are poorly understood. This study leveraged clinical samples from a well-characterized cohort of children hospitalized with COVID-19 or MIS-C to compare immune-mediated biomarkers. Our objective was to identify selected immune molecules that could explain, in part, why certain SARS-CoV-2-infected children developed MIS-C. We hypothesized that type-2 helper T cell-mediated inflammation can elicit autoantibodies, which may account for some of the differences observed between the moderate–severe COVID-19 (COVID^+^) and MIS-C cohort. We enumerated blood leukocytes and measured levels of selected serum cytokines, chemokines, antibodies to COVID-19 antigens, and autoantibodies in children presenting to an academic medical center in Connecticut, United States. The neutrophil/lymphocyte and eosinophil/lymphocyte ratios were significantly higher in those in the MIS-C versus COVID^+^ cohort. IgM and IgA, but not IgG antibodies to SARS-CoV-2 receptor binding domain were significantly higher in the MIS-C cohort than the COVID^+^ cohort. The serum levels of certain type-2 cytokines (interleukin (IL)-4, IL-5, IL-6, IL-8, IL-10, IL-13, and IL-33) were significantly higher in children with MIS-C compared to the COVID^+^ and SARS-CoV-2-negative cohorts. IgG autoantibodies to brain antigens and pentraxin were higher in children with MIS-C compared to SARS-CoV-19-negative controls, and children with MIS-C had higher levels of IgG anti-contactin-associated protein-like 2 (caspr2) compared to the COVID^+^ and SARS-CoV-19-negative controls. We speculate that autoimmune responses in certain COVID-19 patients may induce pathophysiological changes that lead to MIS-C. The triggers of autoimmunity and factors accounting for type-2 inflammation require further investigation.

## 1. Introduction

Multisystem Inflammatory Syndrome in Children (MIS-C) is a delayed immune-mediated multisystem pathologic response to Severe Acute Respiratory Syndrome–Coronavirus 2 (SARS-CoV-2) infection in children. This life-threatening complication of COVID-19 has been defined by the CDC as occurring 2–6 weeks after the initial viral infection, with evidence of fever, severe systemic inflammation, and the involvement of at least two organ systems, frequently the mucosal and gastrointestinal tracts [1,2,3]. Although MIS-C is rare, it is a serious condition that may resemble Kawasaki disease and other multisystem inflammatory disorders [4]. The underlying cause and pathophysiological processes leading to MIS-C and other forms of severe COVID-19 in children are poorly understood; genetic and environmental factors have been suggested [5,6,7]. However, it is not known why some cases of COVID-19 recover and others progress to MIS-C.

In the U.S., there were 9499 cases of MIS-C and 79 attributable deaths from May 2020 to July 2023. Of the MIS-C patients, 46% were 5–11 years of age, and 60% were males [8]. MIS-C frequently mimics the clinical presentation of Kawasaki disease with elevated inflammatory biomarkers [9]. Definitionally, multiple organs are involved in MIS-C [10]. For example, as many as 20% of children with MIS-C had neurological manifestations, including headache, irritability, and encephalopathy [11]. With COVID-19, the loss of one or more of the five senses (e.g., smell, taste, and hearing) has been reported [12,13,14], thus suggesting nervous system involvement. The nervous-system effects associated with COVID-19 have been reviewed [15]. Sensory dysfunction in MIS-C is likely caused by neuroinflammation that may lead to the disruption of neuronal viability and neural circuits, encephalitis, and/or signaling to cells of peripheral organs [16,17].

The delayed onset of MIS-C after the initial infection may be due to evolving host immune responses to the products of damaged cells and not the virus itself. Since SARS-CoV-2 leads to the damage of cells in multiple organs, it may give rise to damage-associated molecular patterns (DAMPs), which induce the production of inflammatory cytokines and chemokines [18,19]. In this context, we hypothesized that children who progress to MIS-C develop a stereotypical type of T helper-cell inflammatory response, which in turn elicits the release of DAMPs and induction of harmful autoantibodies (autoAbs). We speculate that this type of dysregulated immune response may contribute to the delayed symptoms and multiorgan pathology observed in MIS-C.

## 2. Materials and Methods

### 2.1. Study Design and Biological Samples

Biospecimens were obtained from subjects enrolled at Connecticut Children’s, an academic pediatric hospital, between 1 April 2020 and 1 June 2022, in a prospective observational study. After obtaining IRB approval (#21-004, 27 January 2021), participants, birth to ≤21 years of age, were prospectively identified and enrolled into one of three cohorts: (1) patients positive for SARS-CoV-2 infection per antigen (Ag) or PCR testing and hospitalized for moderate-to-severe COVID-19 symptoms and signs (COVID^+^), (2) patients hospitalized and meeting the CDC criteria for MIS-C, or (3) SARS-CoV-2-negative patients undergoing routine ambulatory surgery for conditions unrelated to COVID-19 or an inflammatory disorder (control). Controls were age-matched (±6 months) at a 1:2 ratio to participants in the COVID^+^ and MIS-C cohorts. To ensure correct group assignment, cases were reviewed by three pediatric specialists, and the primary diagnosis was confirmed via a consensus. In addition to case adjudication, controls enrolled in the ambulatory setting with an inflammatory marker of more than three standard deviations above average were excluded to mitigate potential misclassification of participants who may have had an undiagnosed inflammatory condition (*n* = 6). Blood for research purposes was collected at enrollment, and the serum was separated, aliquoted into 200 µL cryovials, and stored at −80 °C until utilized. Demographic, clinical, and laboratory data, including the results from tests ordered within 24 h of admission as part of routine care, were systematically collated. Results were considered abnormal if C-reactive protein (CRP) was ≥0.5 mg/dL, troponin ≥ 0.3 ng/mL, and aminoterminal pro B-type natriuretic peptide (ProBNP) ≥ 1000 pg/mL (125–999 pg/mL was considered indeterminate). Throughout the study period, all patients in the control cohort were tested for SARS-CoV-2 before elective surgery and were negative for acute infection (both COVID and other viral infections) as they were coming in for outpatient surgery. However, they were not excluded if they had a prior history of COVID-19. In addition, COVID-19 vaccines have been deployed since December 2020; therefore, some controls may have been vaccinated. All radiographs and laboratory results were considered by the expert review committee when making cohort group determinations. Thus, if a subject had a co-infection, such as presumed bacterial pneumonia based on an indicative chest CT, a documented bacterial UTI, or bacteremia, that individual would have been excluded from the study. Controls were negative for acute infection (both COVID and other viral infections) as they were coming in for outpatient surgery. However, they were not excluded if they had a prior history of COVID-19. In addition, COVID-19 vaccines have been deployed since December 2020.

### 2.2. Luminex Multiplex Assay

Cytokine/chemokine levels in the serum samples obtained during the initial admission were measured in duplicate using the Luminex^®^ 200^™^ instrument and Milliplex^®^MAP kits (Cat #HSTCMAG-28SK, HCYTA-60K, HCYP2MAG-62K, and HCYP4MAG-64K, EMD Millipore, Burlington, MA 01803, USA), according to the manufacturer’s protocol. A 96-well plate provided with the kit was first washed with 200 µL of wash buffer. The wash buffer was then discarded, and 25 µL of serum sample was added to the plate in duplicate, along with 50 µL of standards and controls which were provided with the kits. Assay buffer (25 µL) was then added to the sample wells, followed by 25 µL of premixed magnetic beads (provided with the kit) to each well. The plate was covered with a dark lid and placed on a plate shaker (200 rpm on a Barnstead 4625 Titer plate shaker) overnight at 4 °C in a dark room. The following day, the plate was washed three times with 200 µL of wash buffer, using a BioTek ELx405^™^ microplate washer with magnetic capture (Agilent Technologies, Inc., Santa Clara, CA 95051, USA). Following washing, detection antibodies (50 µL) were added to each well, and the plate was incubated for 1 h on a plate shaker covered with foil. Streptavidin–phycoerythrin (50 µL) was added to each well, and the plate was incubated for 30 min on a plate shaker covered with foil. Lastly, the plate was washed three times with 200 µL of wash buffer using a BioTek ELx405^™^ microplate washer and analyzed using the Luminex^®^ 200^™^ instrument (calibrated each week with Luminex 200 Calibration and Performance Verification kits: Cat # LX2R-CAL-K25, LX2R-PVER-K25) with 150 µL of sheath fluid present in each well. Standard curves were generated using the Luminex xPONENT^®^ software version 4.3 (Luminex, Austin, TX 78727, USA), and the concentrations of cytokines/chemokines in the serum samples were calculated using the standard curves in pg/mL.

### 2.3. SARS-CoV-2 Antibody (Ab) Microsphere Immunofluorescence Assay

Levels of immunoglobulin (Ig) G1, IgG2, IgG3, IgG4, IgM, and IgA Abs against SARS-CoV-2 nucleocapsid or spike Ags were measured in the serum samples using a microsphere immunofluorescence assay [20]. This is a multiplexed assay based on a Luminex platform, which simultaneously detects Abs to different antigens (Ags) coupled to unique microspheres. Ag-coupled microspheres possess unique fluorescent signatures and are identifiable within the same reaction, while Abs bound to the individual Ags are detected using an anti-human immunoglobulin (anti-isotype) reagent, which has been conjugated to phycoerythrin; this assay has FDA and NY State Clinical Laboratory Evaluation Program and emergency use authorization (CLEP EUA) status for testing, and the assay is performed as follows. A Multiscreen^HTS^ BV Filter Plate (Cat #MSBVN1250, MilliporeSigma, Burlington, MA 01803, USA) was first blocked with 100 µL of phosphate-buffered saline (PBS) with 1% bovine serum albumin (BSA) and 0.05% sodium azide, pH 7.4 (PBN), for two minutes. The contents were discarded, and 190 µL wash buffer (PBS, 0.05% Tween 20, pH 7.4) was added. Contents were aspirated from the wells using a multiscreen resistant pump/manifold, and 25 µL of diluted controls and samples (1:100 dilution in PBN) was added to the appropriate wells. The plate was covered and incubated for 30 min at 37 °C, while shaking, and then washed three times with 190 µL of wash buffer. Then, 50 µL of the diluted secondary Ab of choice was added, resulting in a final concentration of 2 µg/mL. The plate was covered and incubated for 30 min at 37 °C, then washed twice with 190 µL of wash buffer, resuspended in 125 µL of PBN, and mixed with a multichannel pipette to ensure even suspension of the beads. The contents were transferred into a fresh flat-bottom Costar 96-well plate, and median fluorescence intensity (MFI) was measured on a FlexMap 3D^TM^ instrument (bio-techne, Minneapolis, MN 55413, USA).

### 2.4. ELISA for Total Serum IgG

Since some subjects received intravenous immunoglobulin (IVIg) treatment before blood collection, total serum IgG levels were assayed to adjust the autoAb IgG level for individual subjects in proportion to their total IgG levels, calculated as autoAb IgG per mg of total IgG. To measure total serum IgG levels, a flat bottom 96-well Costar assay plate (Cat # 3369, MilliporeSigma, Burlington, MA 01803, USA) was coated with Fab-specific anti-human IgG (2 µg/well, Cat # 15260, MilliporeSigma) and incubated overnight at 4 °C. The next day, the plate was washed three times with PBS with 0.05% Tween-20 (PBST), blocked with 200 µL of PBS-T with 3% BSA for two hours at room temperature, and then it was washed again three times with PBS-T. Then, 100 µL/well of serial dilutions of human IgG from 100 ng/mL (Cat # 14506, MilliporeSigma) or human serum samples (1:160,000 in PBS-T with 1% BSA) was added. The plate was incubated for two hours at room temperature and then washed six times with PBS-T. HRP–anti-human Fc-specific IgG (100 µL/well of 1:40,000 dilution, Cat #A6029, MilliporeSigma) was added to each well and incubated for one hour at room temperature. Lastly, the plate was washed six times with PBS-T, and 100 µL of TMB substrate (3,3′,5,5′-Tetramethylbenzidine, Cat #T4444, Millipore Sigma) was added to each well for color development; absorbance was measured at 450 nm, using an ELISA analyzer (BioTek EL808, Agilent).

### 2.5. ELISA for IgG Anti-Brain autoAbs

A flat-bottom 96-well Costar assay plate (Cat # 3369, Corning) was coated with human-brain whole-tissue lysate (NB820-59177, Novus Biologicals, Centennial, CO 80112, USA) at 10 µg/well and incubated at 4 °C overnight. The next day, the plate was washed three times with PBS-T and blocked with 3% BSA-PBS-T (200 µL) for two hours at room temperature. After washing the plate three times, human serum samples were diluted 1:100 by adding 1% BSA-PBS-T (100 µL/well) and incubated for another two hours at room temperature. The plate was then washed six times, and 100 µL of HRP-tagged anti-human IgG (1:40,000 dilution, Cat #A6029, Millipore Sigma) was added to each well, and the plate was incubated for two hours at room temperature. Lastly, the plate was washed six times, and 100 µL of TMB substrate (3,3′,5,5′-Tetramethylbenzidine, Cat #T4444, Millipore Sigma) was added to each well for color development. Absorbance at 450 nm was measured using an ELISA analyzer (BioTek EL808). The anti-brain IgG autoAbs in each sample were calculated as a proportion of total serum IgG (IgG autoAb OD_450_/mg total IgG), as some subjects received IVIg therapy before blood collection.

### 2.6. ELISA for IgG Anti-dsDNA autoAbs

An empty flat-bottom 96-well Costar assay plate (Cat # 3369, Corning) was incubated under UV light for 16 h. The plate was then coated with 100 µL poly dA-dT (7.5 µg/mL, Cat #D4522, Millipore Sigma) and incubated overnight at 4 °C. The next day, the plate was washed three times with PBS-T and blocked with 250 µL of 10% fetal bovine serum (FBS) in PBS for two hours at room temperature. The plate was washed three times with PBS-T before adding 100 µL of human serum samples diluted (1:100) in 10% FBS in PBS in the plate and incubated for two hours at room temperature. The plate was then washed six times before adding 100 µL of HRP-tagged anti-human IgG (1:40,000 dilution, Cat #A6029, Millipore Sigma) to each well and incubated for two hours at room temperature. Lastly, the plate was washed six times, and 100 µL of TMB substrate (3,3′,5,5′-Tetramethylbenzidine, Cat #T4444, Millipore Sigma) was added to each well for color development. Absorbance at 450 nm was measured using an ELISA analyzer (BioTek EL808). IgG anti-dsDNA level (OD_450_/mg total serum IgG) was calculated as described for anti-brain autoAbs.

### 2.7. ELISA for IgG Anti-Metallothionein (MT) autoAbs

A flat-bottom 96-well Costar assay plate (Cat # 3369, Corning) was coated with recombinant metallothionein (MT2A made in BL21 cells) at 3 µg/well diluted in coating buffer (0.1 M carbonate buffer, pH 9.5) and incubated overnight at 4 °C. The next day, the plate was washed three times with PBS-T and blocked with 3% BSA-PBS-T (200 µL) for two hours at room temperature. After washing the plate three times, 100 µL of human serum samples (diluted 1:100 in 1% BSA-PBS-T) was added, and the plate was incubated for another two hours at room temperature. The plate was washed six times, and 100 µL of HRP-tagged anti-human IgG (1:50,000 dilution, Cat #A6029, Millipore Sigma) was added to each well, and the plate was incubated for one hour at room temperature. Lastly, the plate was washed six times, and 100 µL of TMB substrate (3,3′,5,5′-Tetramethylbenzidine, Cat #T4444, Millipore Sigma) was added to each well for color development. Absorbance at 450 nm was measured using an ELISA analyzer (BioTek EL808), and IgG anti-MT autoAb concentrations (OD_450_/mg total serum IgG) were calculated as described for anti-brain autoAbs.

### 2.8. ELISA of IgG autoAbs to Specific Proteins Captured from Brain Lysate

The human-brain lysate described earlier was added to the wells (1 µg/well) of a flat-bottom 96-well Costar assay plate (Cat # 3369, Corning) that was precoated with 2 µg/well of anti-contactin-associated protein-like 2 (anti-caspr2; H-10; Cat# sc-398454, Santa Cruz Biotechnology, Dallas, TX 75220, USA), anti-pentraxin (Cat# MAB78161-100, R&D systems, bio-techne), anti-aquaporin 4 (AQP4) (4/18; Cat# sc-32739, Santa Cruz Biotechnology), or anti-N-methyl-D-aspartate receptor 1 (anti-NMDAR1; Cat# sc-518043, Santa Cruz Biotechnology). Thereafter, the assay was performed as described for the IgG anti-brain autoAb assay.

### 2.9. Statistical Analyses

Analyses were performed using SPSS 28.0.1.1 (IBM Corporation, Armonk, NY, USA). The primary aim was to determine differences in cytokines/chemokines, autoAbs, and Ig isotypes for COVID-19 Ags between children diagnosed with COVID-19 (COVID^+^) vs. MIS-C and how these two cohorts differed from the controls. Mann–Whitney U or Fisher’s Exact tests were used for univariate comparisons, and Tukey pairwise comparisons with one-way ANOVA or Fisher–Freeman–Halton were used for multivariable comparisons. The means and standard deviations shown in Figure 1 were performed with Prism 9.4 by GraphPad.

## 3. Results

We measured the differences in serum levels of cytokines/chemokines, Ig isotypes to SARS-CoV-2 Ags, and IgG autoAbs to various Ags among children hospitalized for moderate-to-severe COVID-19 (COVID^+^ cohort), children hospitalized with MIS-C, and age-matched non-COVID-19 controls. The cytokine/chemokine analyses were used to assess the relative predominance of type-1 vs. type-2 inflammation, and Abs to COVID-19 Ags were used to help further differentiate adaptive immunity in patients with COVID^+^ vs. MIS-C. The autoAb analyses helped to assess immunoreactivity to self-targets induced by the virus and identify Ags that may be involved in potential resultant pathologies.

Included were 131 participants (29 COVID^+^, 33 MIS-C, and 69 controls) enrolled at a median age of 11 years (IQR 6–15 years), half of whom were Caucasian (50%), and the majority was non-Hispanic (69%) and male (56%). The median age at the time of enrollment did not significantly differ among the three cohorts. There were non-significant differences in the proportion of males in the MIS-C (58%), COVID^+^ (55%), and control (55%) cohorts and significant differences in race among the three cohorts (Table 1). Almost one-quarter (23%) of participants had one or more co-morbidities at the time of enrollment, the majority of which had asthma (Table 2). No subjects were reported as having cardiovascular, kidney, or liver disease, HIV, systemic lupus erythematosus (SLE), or other rheumatological disorders. Two patients in the COVID^+^ cohort died during the study period, one during hospitalization due to complications of COVID-19, and one shortly after discharge, possibly due to complications of COVID-19.

Within the COVID^+^ and MIS-C cohorts, blood was collected for the serum analyses a median of two days after admission, with interquartile ranges of 1–2 days for COVID^+^ and 1.5–4 days for MIS-C (*p* = 0.036). There was no significant difference in the interval between the onset of symptoms and collection of blood samples for these groups (*p* = 0.22), with interquartile ranges of 3–10 days for COVID^+^ and 4.5–9 days for MIS-C. The symptoms present at the time of admission for patients in the COVID^+^ and MIS-C cohorts were significantly different between the two cohorts (Table 3). However, unlike the other symptoms, a greater proportion of the COVID^+^ cohort had upper and lower respiratory symptoms compared to the MIS-C cohort. Most results from laboratory tests ordered within 24 h of admission showed significant differences between the COVID^+^ and MIS-C cohorts, including the neutrophil/lymphocyte ratio (NLR) and the eosinophil/lymphocyte ratio (ELR) (*p* < 0.001) (Table 4). Not unexpectedly, as shown in Table 5, the MIS-C cohort had a longer length of stay and underwent more interventions than the COVID^+^ cohort.

Typically, type-1 and type-2 cytokine and chemokine responses reflect defenses against viral infections and parasitic infections, respectively. Surprisingly, interferon (IFN)-γ measured in the sera of COVID^+^ and MIS-C cohorts did not differ from the control cohort; however, the serum levels of IFNα2 were higher in MIS-C than in the COVID^+^ and control cohorts, and the serum levels of IFN-β were higher in MIS-C than in the control cohort (Table 6). IL-12 (p70) and GM-CSF, which also are usually grouped with type-1 immune responses, also did not significantly differ among the cohorts. IL-18 was higher in MIS-C than in the COVID^+^ and control cohorts, but it did not promote an IFN-γ response as would be expected, since it typically induces IL-12 [21]. Notably, most of the type-2 cytokines (IL-4, IL-5, IL-6, IL-13, IL-21, and IL-33) were significantly higher in the MIS-C cohort than in the COVID^+^ and control cohorts. The COVID^+^ and MIS-C cohorts had higher levels of IL-15 than the control cohort (*p* < 0.001), but there was no significant difference between the COVID^+^ and MIS-C cohorts. The anti-inflammatory cytokine, IL-10, usually increases to control inflammation, and as expected, the MIS-C cohort had a higher level than the COVID^+^ and control cohorts. The pro-inflammatory cytokine IL-17A’s level was not different between the three cohorts. The chemokines CCL3, CCL4, and CXCL11, which affect innate cell activation and recruitment, were significantly higher in the serum of the MIS-C cohort than in the COVID^+^ and control cohorts.

Based on the significantly higher incidence of skin rashes occurring with MIS-C (Table 3) and the higher levels of cytokines (IL-5, IL-8, IL-13, and IL-33; Table 6) known to influence eosinophil activity [22,23], we performed a further evaluation of eosinophils and ELR. The median numbers of eosinophils and ELR were both significantly higher in the MIS-C cohort than in the COVID^+^ cohort (eosinophils 1.9 (IQR 0–8.1) vs. 0.67 (IQR 0–4.8), *p* = 0.004; ELR 0.31 (IQR 0.02–0.34) vs. 0.004 (IQR 0.00–0.037), *p* < 0.001, respectively). We did not collect clinical information about atopy, or drug or environmental allergies, as it was not part of the original protocol. Although the absolute number of eosinophils did not correlate with any of the serum cytokine/chemokine levels, within the MIS-C group, ELR levels were significantly higher in subjects with skin rash (median, 0.13 (IQR, 0.04–0.37)) compared with those who did not have skin rash at the time of admission (median, 0.01 (0.00–0.12); *p* = 0.002).

Given that treatment for MIS-C patients included IVIg to mitigate inflammation, we attempted to determine if the use of IVIg had a confounding effect on our biomarker analyses. We assessed the serum cytokine and chemokine levels of subjects with MIS-C who had their blood collected before versus after IVIg treatment (Table 7). Serum was collected from six children (18%) before IVIg and 27 (82%) after IVIg treatment. The levels of the Th1 cytokine IFN-γ and the chemokine CCL20 (macrophage inflammatory protein-3) often associated with IFN-γ were non-significantly lower in serum collected after IVIg. Surprisingly, the Th2 cytokines IL-4, IL-5, IL-8, and IL-13, as well as IL-1β, were higher in the serum collected after IVIg.

For the Ig isotypes (IgA, IgM, and IgG1-IgG4) directed against three antigenic determinants of SARS-CoV-2 (spike receptor binding domain (RBD), nucleocapsid (NC), and Spike 1), the median IgG, IgM, and IgA anti-RBD and Spike 1 levels were highest in the MIS-C patients (Table 8). However, the difference in levels between MIS-C and COVID were significant only for IgM and IgA, presumably because of the small sample size and distribution of outliers for IgG. Surprisingly, the IgG1 anti-RBD concentrations were highest in the control cohort, but the relatively lower levels of IgG1 in the MIS-C group likely reflected a shift of the immune response in that group to other IgG subtypes (Table 8).

To ensure valid comparisons among cohorts, autoAb levels were normalized to the total IgG in each serum sample. Due to limitations in the availability of sera for the autoAb assessments, the number of samples analyzed varied across cohorts. Nevertheless, some autoAb associations were still apparent. After adjustment, sera from the MIS-C cohort had a higher level of IgG to caspr2 than the COVID^+^ and control cohorts (Table 9). The MIS-C cohort also had significantly higher levels of IgG to brain Ags and pentraxin than the control cohort. Although not significant, IgG anti-brain and pentraxin were higher in the MIS-C cohort vs. subjects diagnosed with COVID^+^. There were no significant differences in anti-AQP4, anti-dsDNA, anti-MT, or anti-NMADR1 levels among cohorts (Table 9). It appears that the levels of IgG autoAbs to brain Ags, caspr2, and pentraxin in MIS-C patients segregate and potentially define two separate populations of MIS-C patients (Figure 1). The two relatively distinct groups within the MIS-C cohort will need further analyses in future studies to determine if demographics or other clinical features may be important regarding the IgG autoAb levels.
Figure 1Distribution of autoAb levels adjusted for serum IgG concentration for individuals within each cohort. Individual values, and mean and standard deviation are shown. AutoAb levels with significant differences (Table 9) are shown. MIS-C had the greatest variance within a cohort, and there appears to be a bimodal distribution within the MIS-C cohort.
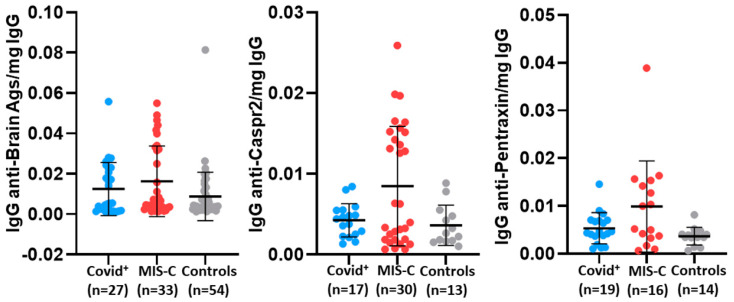


## 4. Discussion

The pathophysiological mechanisms that explain why some children with COVID-19 progress to MIS-C are not well understood. We undertook a prospective study comparing children diagnosed with moderate-to-severe COVID-19 or MIS-C and a healthy control group to test our hypothesis that a dysregulated immune response to the virus, driven by a predominance of type-2 helper T cell-mediated inflammation and production of autoantibodies, could contribute to the development of MIS-C. Our investigation of immune responses revealed that, compared to healthy controls, MIS-C patients had (1) a significant upregulation of several proinflammatory cytokines (CCL3 [MIP-1α], CCL4 [MIP-1β], CXCL11, and tumor necrosis factor (TNF)-α) that are known to recruit and activate neutrophils, monocytes, natural killer cells, and T cells; (2) a broad shift of CD4^+^ T cell-mediated immune responses toward type-2 helper T (Th2) cell cytokines (IL-4, IL-5, IL-6, IL-13, and IL-33) that can mediate B-cell stimulation and differentiation and upregulate antibody production; and (3) an increase in type I (IFN-α2, IFN-β) but not type II (IFN-γ) immunoregulatory antiviral cytokines. Consistent with our hypothesis, we also observed significantly elevated concentrations of a group of autoAbs (anti-caspr2, anti-brain Ags, and anti-pentraxin) in the MIS-C cohort.

To fight the SARS-CoV-2 infection, it is not surprising that some cytokine/chemokine levels promoting innate immunity such as CCL3, CCL4, CXCL11, and TNF-α would be elevated. MIS-C patients are defined as having a severe delayed hyperinflammatory condition occurring 2–6 weeks after antecedent SARS-CoV-2 infection [9,24,25]. The inflammatory biomarkers remained higher in the MIS-C cohort than the COVID^+^ cohort, even though, at 2–6 weeks post-infection, the MIS-C patients should have less virus when samples were obtained. The higher amount of IL-4, IL-5, IL-6, IL-13, and IL-21 in the MIS-C sera would explain the higher amount of Abs. IL-4, IL-5, and IL-6 were originally referred to as B cell-stimulatory, -growth, and -differentiation factors, respectively, reflecting their important role in humoral responses. IL-13’s functions are very similar to IL-4; both cytokines have a common receptor (IL-4Rα) for signaling cells. IL-21 is known to support follicular helper T (Tfh) cell development; Ig affinity maturation; plasma cell differentiation, which is promoted by Tfh cells; and B-cell memory responses [22,26,27]. The alarmin cytokine IL-33 of the IL-1 family is also higher in the MIS-C than the COVID^+^ cohort, and it promotes Th2, as well as innate lymphoid cell type-2 (ILC2) activity [28]. Like DAMPs, IL-33 is released from many different types of damaged cells, and it plays an important role in allergic and non-allergic inflammation [29]. The role of IL-10 in type-1 and type-2 inflammation, as well as COVID-19, remains to be determined [30,31,32]. However, IL-10 is often considered an anti-inflammatory cytokine [33], and expression increases to regulate inflammation [34] and augment a Th2-type response [35]. The shift away from Th1 and toward Th2 responses has been shown with HIV infection and loss of the antioxidant glutathione (GSH) [36,37], and GSH depletion has been implicated in COVID-19 mortality and pathophysiology [38]. GSH is needed to aid Th1 more than Th2 responses [39,40] and to provide metabolic support for T-cell effector functions against pathogens, especially viruses [41]. However, prior depletion of GSH can increase oxidative stress and inflammation due to increased cell damage [42,43].

As expected, children with COVID-19 were more likely to present with upper and lower respiratory symptoms. On the other hand, those with MIS-C had significantly higher rates of fever; mucocutaneous, gastrointestinal, and non-specific systemic symptoms that were associated with significantly higher levels of neutrophil counts, CRP, and ProBNP; and significantly lower hemoglobin concentration and lymphocyte and platelet counts. Although 27 subjects had blood drawn after administration of IVIg, levels of IFN-γ were not significantly different from those of the six subjects who were tested before IVIg. On the other hand, Th2 cytokines (IL-4, IL-5, IL-8, and IL-13), as well as IL-1β, were higher in serum collected after IVIg, but that may have been confounded by delayed sampling with natural progression of the intrinsic immune response. Overall, the duration of symptoms before presentation and before testing were similar between the COVID^+^ and MIS-C groups.

In contrast to our findings, Klocperk et al. [44] reported elevated levels of IFN-γ but not IFN-α; autoreactive antibodies such as anti-Ro60, anti-Sm antigen and anti-extractable nuclear antigen were also reported. On the other hand, Dorio et al. [45] described IFN-γ dysregulation with distinct populations with either high or low levels of IFN-γ, where patients with higher levels had less severe MIS-C. They concluded that IFN-γ heterogeneity may provide clinically relevant insights into disease pathogenesis. Other investigators have previously published comparisons of various IL levels in MIS-C vs. severe COVID [46], although not as many types as in the current study. However, we found significantly higher IL levels of several types in MIS-C cases (Table 6), and this finding was not observed previously.

Surprisingly, the IgG1 serum levels to the COVID-19 RBD epitope were highest in the control cohort compared to either the MIS-C or COVID^+^ cohort. COVID-19 vaccines have been deployed since December 2020; therefore, elevated IgG1 anti-RBD antibodies could be explained by previous vaccination or possibly a prior mild unreported COVID-19 infection. Thus, a vaccination or previous infections, along with possibly fewer lymphocytes in the COVID^+^ cohort or less B-cell activation to some viral Ags in the COVID^+^ cohort could explain the difference; lymphoid subpopulations were not assessed. The higher median IgM, IgA, and IgG1 values in the sera of the MIS-C cohort to some of the COVID-19 Ags may relate to the higher type-2 cytokines (IL-4, IL-5, IL-6, and IL-13) and IL-21 enhanced Tfh activity known to promote B-cell activities, as mentioned earlier. The greater promotion of an IgA anti-RBD response in the MIS-C cohort compared to the COVID^+^ and control cohorts may relate to a greater influence of the gut microbiota on mucosal immunity. The gut microbiome is modified by COVID-19 [47], and differences in composition have been reported between MIS-C and COVID^+Yes^ and controls [48]. IgA responses are related to mucosal immunity [49], and IgA may be elevated to lessen any leakage of the virus from the gut to aid any loss of the mucus layer [50]. As noted in Table 3, the MIS-C cohort does have more gastrointestinal disturbances than the COVID^+^ cohort, suggesting disturbances of microbiome–gut homeostasis.

A higher NLR value has been linked to attenuated immune defenses against microbes due to greater oxidative stress and cell damage [51,52,53]. As previously demonstrated [54], our MIS-C patients had a significantly higher median NLR of 10.14 compared to 3.28 for the COVID^+^ cohort (Table 4). An NLR > 6.5 has been useful in predicting greater disease severity and mortality in COVID-19 [55,56] and MIS-C patients [54]. IL-8 (CXCL8) is a chemokine for granulocytes and is an activator of neutrophils [57]. IL-8, which was significantly higher in MIS-C vs. the COVID^+^ and control groups, may have upregulated neutrophil counts and contributed to oxidative stress and cell damage. Another hematological difference was higher eosinophil numbers in the MIS-C cohort. Elevated eosinophils are associated with more Th2 and type-2 innate lymphoid cell activity, as well as atopic responses [58]. As shown in Table 4, the MIS-C cohort had more eosinophils than the COVID^+^ patients. IL-33 induces eosinophils to release IL-13 [59], and both were elevated in the MIS-C cohort. Like NLR, our MIS-C patients had a significantly higher median ELR of 0.132 compared to 0.004 for the COVID^+^ cohort. The type-2 cytokines eliciting eosinophils [60] and the skin rashes with MIS-C may be like Stevens–Johnson Syndrome or an atopic response with the presence of eosinophils [61,62,63].

Due to the higher levels of IgG anti-brain Ags and the IgG anti-caspr2, which has been associated with encephalitis [64], we considered some neurologic sequelae associated with MIS-C patients. However, the new definition of MIS-C does not include neurologic symptoms [65]. The involvement of autoimmune problems associated with MIS-C remains an issue not yet resolved. MIS-C patients also had anti-pentraxin, which has been associated with autoimmune disease [66]. There have been reports connecting MIS-C with autoimmune responses [67,68], although it remains controversial [69].

Although females usually have a higher prevalence of autoimmune diseases than males, males may have similar or slightly higher rates of psoriasis, ulcerative colitis, and type-1 diabetes (T1D) [70]. The percentage of males developing MIS-C is slightly higher than that of females [71]. The microbiome–gut–brain axis affects immunity and inflammation [72] and has been implicated in neural disorders [73,74,75], T1D [76], and MIS-C [77,78]. The microbiome also affects autoimmune diseases [79]. Since the microbiome–gut–brain axis affects immunity, autoAb production, and systemic inflammation as occurs in MIS-C, the autoAb specificities associated with autoimmune disorders may be informative for the mechanisms affecting the pathophysiology of MIS-C. COVID-19 has been reported to have some connections with autoimmunity [80,81,82,83,84]. Some of the more detrimental outcomes of MIS-C may be due to Abs to self-constituents. AutoAbs to NMDAR, AQP4, caspra2, and other brain Ags are associated with neurological disease [85]; 10 of 19 patients with anti-caspr2 had neuromyotonia or Morvan’s syndrome [86]. AutoAbs to brain Ags are being considered as initiators of some psychosis [87], and COVID-19 has been associated with onset of some psychosis [88,89]. AutoAb specificities and DAMPs from cells damaged by the virus and higher type-2 inflammation need further consideration and investigation. Although the levels of IgG to brain Ags, caspr2, and pentraxin in the MIS-C cohort appear to be bimodal, there was no clinical or biomarker that could explain this differential.

The limitations of the study include the enrollment of subjects at a single center, which may limit generalizability, and, although appropriately adjusted, analyses were performed with multiple comparisons with a limited sample size. Additionally, it must be noted that the children enrolled in our study do not necessarily reflect the makeup of all admitted children with MIS-C and COVID, but rather a selected population of enrollees. We had a near-universal approach to children suspected of having MIS-C due to its rarity. However, given the overwhelming number of children admitted with acute COVID-19 at times, not all were approached for enrollment. Additionally, well-described differences in racial/ethnic clinical trial enrollments in pediatric research may also explain discrepancies in our enrollments. The control group in our study was 70% White, while the two hospitalized cohorts had significantly fewer Whites, with only 15% of MIS-C and 41% of COVID^+^ patients. This discrepancy reflects the patient demographics of our inpatient and outpatient clinics in Connecticut. Our outpatient and surgical subspecialties draw patients from across the state, whereas our inpatient unit primarily serves Hartford County, which has a more diverse racial/ethnic makeup. Similar findings were published in 2024, where an outpatient mild-COVID cohort was predominantly White [90]. Multiple potential explanations exist for the differences observed in the hospitalized cohorts (COVID and MIS-C). Both conditions have known racial/ethnic disparities in their distribution. According to Vicetti Miguel et al. [91], racial/ethnic differences in the incidence of COVID-19 and MIS-C are well-documented. Encinosa et al. [92] found that MIS-C incidence varies twofold by race, with 0.97 per 100,000 in White children compared to 1.99 per 100,000 in non-Hispanic Black children. Similarly, COVID-19 incidence varied from 4.4 per 100,000 in White children to 6.6 per 100,000 in Black children.

In summary, the cytokine/chemokine serum levels within the MIS-C cohort appear to upregulate certain IgG autoAb levels, which may contribute to the pathogenesis of MIS-C.

## Figures and Tables

**Table 1 viruses-16-00950-t001:** Demographics of participants.

Characteristic	Total Cohort (*n* = 131)	COVID^+^ (*n* = 29)	MIS-C (*n* = 33)	Control (*n* = 69)	*p*-Value
Age (mos) IQR	127 (72–185)	110 (38–196)	116 (70–158)	144 (74–189)	0.39
Age range (mos)	1–260	1–214	2–212	8–260	
Male	80 (56%)	16 (55%)	19 (58%)	38 (55%)	0.97
Race					<0.001
Black	22 (17%)	4 (14%)	13 (39%)	5 (7%)	
White	65 (50%)	12 (41%)	5 (15%)	48 (70%)	
Other Race	26 (20%)	10 (35%)	12 (36%)	4 (6%)	
>1 Race	13 (10%)	1 (3%)	3 (9%)	9 (13%)	
Prefer not to answer	5 (4%)	2 (7%)	-	3 (4%)	
Hispanic	40 (31%)	10 (35%)	12 (36%)	18 (26%)	0.67
Prefer not to answer	2 (2%)	0	0	2 (3%)	

Data are given as *n* (%) or median (IQR and range). Comparisons were made using 1-way ANOVA with Tukey adjusted pairwise comparison for continuous variables, or Fisher–Freeman–Halton exact test for categorical variables with the unadjusted *p*-value reported.

**Table 2 viruses-16-00950-t002:** Comorbidities of participants.

Comorbidity	Total Cohort (*n* = 131)	COVID^+^ (*n* = 29)	MIS-C (*n* = 33)	Control (*n* = 69)
Asthma	26 (18%)	11 (38%)	8 (24%)	7 (9%)
Other chronic lung disease	1 (1%)	1 (3%)	0	0
Hypertension	1 (1%)	1 (3%)	0	0
Immunocompromised	1 (1%)	1 (3%)	0	0
Diabetes	3 (2%)	3 (10%) *	0	0
IBD	1 (1%)	1 (3%)	0	0
JIA	1 (1%)	1 (3%)	0	0
Other rheumatic disease	1 (1%)	1 (3%)	0	0
Cancer	1 (1%)	1 (3%)	0	0
Sickle cell disease	2 (1%)	1 (3%)	1 (3%)	0
Serious mental illness	3 (2%)	1 (3%)	0	2 (3%)

Data are given as *n* (%). * Type I (n = 2) and type II (*n* = 1).

**Table 3 viruses-16-00950-t003:** Symptoms at time of admission.

Symptom	COVID^+^ (*n* = 29)	MIS-C (*n* = 33)	*p*-Value
Fever	16 (55%)	31 (94%)	<0.001
Mucocutaneous ^1^	3 (10%)	20 (61%)	<0.001
Upper respiratory ^2^	20 (69%)	13 (39%)	<0.05
Lower respiratory ^3^	15 (52%)	5 (15%)	<0.005
Gastrointestinal ^4^	17 (59%)	30 (91%)	<0.01
Neurologic ^5^	0	0	
Systemic ^6^	14 (48%)	30 (91%)	<0.001
No symptoms present	0	0	
Days of symptoms prior to admission	3.0 (1.5–8.0)	4.0 (3.0–5.0)	0.36

Data are reported as *n* (%) or median (IQR). Comparisons made using Fisher’s exact test or Mann–Whitney U. ^1^ Skin rash and conjunctivitis; ^2^ sore throat, congestion, and cough; ^3^ shortness of breath, apnea, chest pain, and wheezing; ^4^ nausea/vomiting, diarrhea, and abdominal pain; ^5^ loss of taste and/or smell; ^6^ headache, muscle ache, fatigue, or chills.

**Table 4 viruses-16-00950-t004:** Laboratory results in first 24 h of admission.

Diagnostic Lab	COVID^+^ (*n* = 29)	MIS-C (*n* = 33)	*p*-Value
**CBC ordered**	**25 (86%)**	**31 (94%)**	
WBC (10^3^/µL)	8.5 (4.7–15.4)	9.2 (7.2–14.5)	0.49
Hemoglobin (g/dL)	13.4 (11.8–14.4)	11.8 (10.3–12.7)	0.005
Platelet count (10^3^/µL)	260 (206–354)	180 (121–261)	0.001
Neutrophils (10^3^/µL)	5.6 (2.4–11.3)	7.3 (5.5–11.4)	0.002
Lymphocyte (%)	21.1 (12.6–41.7)	8.0 (5.6–17.2)	<0.001
Lymphocytes (10^3^/µL)	1.8 (1.3–3.2)	0.7 (0.5–1.8)	0.002
Eosinophils (10^3^/µL)	0.67 (0–4.8)	1.9 (0–8.1)	0.004
Neutrophil/lymphocyte ratio	3.28 (1.06–6.36)	10.14 (4.31–16.0)	<0.001
Eosinophil/lymphocyte ratio	0.004 (0–0.037)	0.132 (0.02–0.338)	<0.001
**CRP ordered**	**22 (76%)**	**31 (94%)**	
CRP (mg/dL) median (IQR)	1.16 (0.3–4.89)	15.9 (10.9–21.7)	<0.001
CRP (mg/dL) range	0.1–9.8	2.6–52.2	
CRP elevated ^1^	13 (59%)	31 (91%)	<0.001
**Troponin ordered**	**13 (45%)**	**30 (91%)**	
Troponin I (ng/mL) median (IQR)	0.30 (0.30–0.30)	0.30 (0.30–0.31)	0.49
Troponin I (ng/mL) range	0.30–0.51	0.30–7.67	
Troponin elevated ^2^	13 (100%)	30 (100%)	1.00
**ProBNP ordered**	**13 (45%)**	**30 (91%)**	
ProBNP (pg/mL) median (IQR)	197 (19–1737)	1583 (672–6110)	0.027
ProBNP (pg/mL) range	5–70,000	44–70,000	
ProBNP elevated ^3^	4 (31%)	19 (63%)	0.09
ProBNP indeterminate ^4^	3 (23%)	6 (20%)	1.00

Data are given as median (IQR) or n (%). Comparisons were made using Fisher’s exact test or Mann–Whitney U. Note that some subjects did not have labs ordered. ^1^ CRP ≥ 0.5 mg/dL; ^2^ troponin ≥ 0.3 ng/mL; ^3^ ProBNP ≥ 1000 pg/mL; ^4^ ProBNP 125–999 pg/mL.

**Table 5 viruses-16-00950-t005:** Inpatient-stay details and interventions.

Characteristic	COVID^+^ (*n* = 29)	MIS-C (*n* = 33)	*p*-Value
Length of stay (days)	4 (2–6)	6 (4–9)	<0.05
ICU	9 (31%)	17 (52%)	0.13
Intubation	0	0	
Ventilation (BiPAP)	1 (3%)	0	0.47
Received IVIg	1 (3%)	33 (100%)	<0.001
Received steroids	14 (48%)	32 (97%)	<0.001
Received immunomodulator	0	2 (6%)	0.49
Received antimicrobial	18 (62%)	31 (94%)	<0.005

Data are given as n (%) or median (IQR). Comparisons were made using Fisher’s exact test or Mann–Whitney U test.

**Table 6 viruses-16-00950-t006:** Serum cytokine and chemokine levels in COVID^+^, MIS-C, and COVID- control.

Cytokine/Chemokine	COVID^+^ (*n* = 29)	MIS-C (*n* = 33)	Control (*n* = 69)	COVID^+^ vs. Control *p*-Value	MIS-C vs. Control *p*-Value	COVID^+^ vs. MIS-C *p*-Value
CCL3	28 (18–35)	44 (30–51)	18 (12–27)	0.66	<0.001	0.003
CCL4	33 (20–62)	69 (39–88)	35 (24–49)	0.29	<0.001	0.031
CCL20	31 (24–36)	41 (35–55)	30 (26–40)	0.67	0.034	0.37
CX3CL1	219 (185–266)	255 (221–344)	241 (194–300)	0.15	0.49	0.78
CXCL11	107 (51–346)	1010 (300–1458)	53 (40–71)	0.38	<0.001	<0.001
GM-CSF	83 (33–165)	110 (65–273)	72 (36–152)	0.89	0.12	0.10
IFN-α2	83 (8–150)	148 (60–236)	96 (26–190)	0.69	0.035	0.016
IFN-β	488 (488–488)	488 (478–1068)	488 (488–488)	0.98	0.031	0.12
IFN-γ	77 (46–149)	113 (78–189)	76 (49–156)	0.86	0.88	0.67
TNF-α	15 (11–20)	28 (19–47)	14 (11–18)	0.80	<0.001	<0.001
IL-1β	3.2 (2.1–4.5)	4.8 (3.4–6.2)	3.1 (2.1–5.4)	0.53	0.23	0.06
IL-2	5.2 (2.8–79)	6.9 (4.2–8.0)	4.3 (2.7–6.4)	0.54	0.020	0.38
IL-4	39 (19–115)	686 (328–953)	49 (26–194)	0.80	<0.001	<0.001
IL-5	4.6 (2.6–7.8)	21 (11–26)	4.9 (2.6–8.6)	0.99	<0.001	<0.001
IL-6	13 (5–24)	73 (47–101)	4.4 (1.1–11.8)	0.30	<0.001	<0.001
IL-7	16 (11–19)	18 (14–23)	13 (11–17)	0.96	0.016	0.09
IL-8	18 (9–47)	112 (58–152)	9.4 (5.1–22.8)	0.22	<0.001	<0.001
IL-10	35 (22–41)	79 (42–176)	13 (8–20)	0.58	<0.001	0.002
IL-12(p70)	4.9 (3.0–6.5)	4.6 (3.1–6.9)	4.8 (3.4–7.5)	0.31	0.35	0.99
IL-13	7.9 (4.2–30.4)	127 (38–179)	8.6 (2.8–37.5)	0.95	<0.001	<0.001
IL-15	5.9 (3.0–19.5)	24 (14–35)	3.0 (3.0–8.3)	<0.001	<0.001	0.40
IL-17a	16 (11–25)	22 (14–30)	16 (9–28)	0.88	0.73	0.54
IL-18	218 (93–417)	652 (438–1041)	103 (46–217)	0.23	<0.001	<0.001
IL-21	5.8 (3.8–9.7)	9.1 (7.3–12.6)	5.8 (3.9–8.8)	0.99	0.006	0.038
IL-23	487 (188–1316)	966 (748–1541)	591 (311–1126)	0.99	0.19	0.37
IL-33	23 (20–286)	464 (317–670)	32 (20–151)	0.67	<0.001	0.023

Data are given as median (IQR). Comparisons were made using 1-way ANOVA with Tukey-adjusted pairwise comparison.

**Table 7 viruses-16-00950-t007:** Serum cytokine/chemokine levels before or after IVIg treatment of the MIS-C patients.

Cytokine/Chemokine	Serum Collected before IVIg (*n* = 6)	Serum Collected after IVIg (*n* = 27)	*p*-Value
CCL3	30 (13–56)	45 (33–49)	0.17
CCL4	66 (31–101)	69 (44–85)	0.95
CCL20	70 (47–163)	38 (31–48)	<0.05
CX3CL1	250 (226–358)	255 (197–344)	0.91
CXCL11	1030 (454–1499)	969 (293–1498)	0.77
GM-CSF	143 (59–287)	105 (66–270)	0.73
IFN-α2	63 (29–348)	156 (68–221)	0.30
IFN-β	493 (488–7314)	488 (441–1054)	0.30
IFN-γ	216 (86–304)	112 (56–163)	0.08
TNF-α	41 (23–54)	24 (19–48)	0.35
IL-1β	4.0 (2.8–4.6)	5.2 (3.8–6.3)	<0.05
IL-2	7.2 (4.4–8.1)	6.9 (3.7–8.1)	0.95
IL-4	40 (14–593)	810 (544–974)	<0.05
IL-5	6.9 (2.5–16.0)	23 (16–27)	<0.05
IL-6	45 (25–151)	74 (55–96)	0.51
IL-7	17 (15–23)	18 (14–22)	0.84
IL-8	28 (13–99)	119 (81–165)	<0.05
IL-10	198 (166–810)	64 (36–102)	<0.005
IL-12(p70)	5.1 (3.7–6.7)	4.4 (2.8–6.9)	0.77
IL-13	12 (3–63)	154 (49–190)	0.001
IL-15	39 (16–60)	24 (12–34)	0.06
IL-17a	28 (17–45)	19 (13–28)	0.28
IL-18	977 (545–1508)	630 (429–895)	0.24
IL-21	8.3 (6.5–13.6)	9.5 (7.3–12.1)	0.67
IL-23	879 (635–1501)	1016 (726–1621)	0.57
IL-33	39 (20–1069)	554 (349–692)	0.06

Data are given as median (IQR). Between-group comparisons were made using Mann–Whitney U.

**Table 8 viruses-16-00950-t008:** Antibodies to COVID antigens.

Antibody Isotype	COVID^+^ (*n* = 17)	MIS-C (*n* = 26)	Control (*n* = 55)	COVID^+^ vs. Control *p*-Value	MIS-C vs. Control *p*-Value	COVID^+^ vs. MIS-C *p*-Value
IgM RBD	0.58 (0.20–4.36)	0.90 (0.25–2.05)	0.20 (0.14–0.29)	<0.001	0.30	0.034
IgM NC	0.14 (0.07–0.32)	0.13 (0.08–0.42)	0.11 (0.06–0.21)	0.017	0.62	0.18
IgM Spike1	0.36 (0.22–13.35)	2.53 (0.54–4.67)	0.29 (0.15–0.42)	<0.001	0.60	0.009
IgA RBD	0.21 (0.04–2.79)	3.34 (0.92–8.14)	0.41 (0.08–1.92)	1.00	0.002	0.024
IgA NC	0.10 (0.04–0.53)	0.29 (0.14–0.49)	0.13 (0.08–0.41)	0.94	0.08	0.37
IgA Spike1	0.26 (0.12–7.93)	13.1 (3.1–30.8)	1.98 (0.18–8.80)	0.81	0.08	0.07
IgG1 RBD	6.09 (0.02–21.29)	34.8 (27.7–51.9)	103.2 (1.0–196.3)	<0.001	0.008	0.55
IgG1 NC	4.18 (0.02–13.53)	31.7 (16.7–67.0)	6.98 (0.08–70.1)	0.11	0.88	0.08
IgG1 Spike1	17.93 (0.07–85.12)	348.0 (178.5–512.3)	521.5 (2.0–1164.5)	<0.001	0.10	0.21
IgG2 RBD	0.23 (0.04–0.66)	1.22 (0.81–1.65)	0.49 (0.15–1.81)	0.29	0.59	0.82
IgG2 NC	0.53 (0.14–1.67)	2.21 (1.51–8.01)	0.70 (0.28–4.15)	0.71	0.13	0.70
IgG2 Spike1	0.50 (0.14–2.28)	4.28 (2.34–6.88)	2.05 (0.32–8.58)	0.36	0.60	0.88
IgG3 RBD	0.11 (0.03–0.90)	1.34 (0.85–3.00)	0.93 (0.36–3.48)	0.61	0.50	1.00
IgG3 NC	0.38 (0.02–1.78)	4.4 (2.3–14.3)	0.91 (0.45–19.5)	0.15	0.54	0.67
IgG3 Spike1	0.27 (0.07–4.57)	3.7 (2.2–6.4)	2.40 (0.40–8.93)	0.57	0.41	0.99
IgG4 RBD	0.38 (0.38–0.69)	0.75 (0.47–1.03)	1.75 (0.50–29.4)	0.18	0.06	0.99
IgG4 NC	0.38 (0.13–0.69)	1.13 (0.55–2.24)	0.64 (0.57–0.86)	0.94	0.03	0.07
IgG4 Spike1	0.25 (0.25–0.56)	0.50 (0.25–0.80)	0.94 (0.56–25.4)	0.16	0.06	0.99

Data are given as median (IQR). RBD = Spike Receptor Binding Domain; NC = nucleocapsid. Comparisons were made using 1-way ANOVA with Tukey-adjusted pairwise comparison.

**Table 9 viruses-16-00950-t009:** Serum AutoAb levels.

AutoAb	COVID^+^	MIS-C	Control	COVID^+^ vs. Control *p*-Value	MIS-C vs. Control *p*-Value	COVID^+^ vs. MIS-C *p*-Value
**Anti-dsDNA**	***n* = 27**	***n =* 29**	***n =* 20**			
Median (IQR)	0.008 (0.005–0.019)	0.011 (0.009–0.032)	0.008 (0.004–0.018)	0.92	0.15	0.24
**Anti-brain**	***n =* 27**	***n* = 33**	***n* = 54**			
Median (IQR)	0.005 (0.002–0.023)	0.007 (0.003–0.033)	0.004 (0.003–0.010)	0.52	0.045	0.54
**Anti-AQP4**	***n* = 13**	***n* = 12**	***n* = 16**			
Median (IQR)	0.002 (0.001–0.005)	0.002 (0.001–0.004)	0.002 (0.001–0.003)	0.77	0.69	0.99
**Anti-Caspr2**	***n* = 17**	***n* = 30**	***n* = 13**			
Median (IQR)	0.004 (0.002–0.005)	0.005 (0.005–0.015)	0.003 (0.002–0.005)	0.95	0.028	0.038
**Anti-MT**	***n* = 27**	***n* = 29**	***n* = 20**			
Median (IQR)	0.012 (0.005–0.021)	0.007 (0.005–0.042)	0.010 (0.006–0.030)	0.58	0.96	0.35
**Anti-pentraxin**	***n* = 19**	***n* = 16**	***n* = 14**			
Median (IQR)	0.005 (0.004–0.007)	0.008 (0.003–0.015)	0.004 (0.003–0.004)	0.71	0.016	0.07
**Anti-NMDAR1**	***n* = 13**	***n* = 12**	***n* = 16**			
Median (IQR)	0.002 (0.001–0.003)	0.002 (0.001–0.002)	0.001 (0.001–0.002)	0.07	0.78	0.32

Data are given as median (IQR). Comparisons were made using 1-way ANOVA with Tukey-adjusted pairwise comparison.

## Data Availability

The original contributions presented in the study are included in the article; further inquiries can be directed to the corresponding author.

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
