# Peer review of "Inflammatory and Autoimmune Aspects of Multisystem Inflammatory Syndrome in Children (MIS-C): A Prospective Cohort Study"

_viruses, 2024, doi:10.3390/v16060950_

Round 1
Reviewer 1 Report
Comments and Suggestions for Authors
V The authors reported inflammatory and autoimmune manifestations of MIS-C in children.
This study aims to explain why certain SARDS-CoV-2-infected children developed MIS-C.
They speculated that Th2-mediated inflammation can elicit autoantibodies, which may account for some of the differences observed between moderate-severe COVID-19 and MIS-C cohort.
Critics:
*Curiously, in the Ig isotypes (IgA, IgM, and IgG1-IgG4) directed against three antigenic determinants of SARS-CoV-2 (spike receptor binding domain [RBD], nucleocapsid [NC], and Spike 1), the IgG1 anti-RBD, and some other results were high in the control cohort. Assuming that controls were negative for COVID-19, the results must be explained more clearly.
*In the peripheral blood examination, such as the neutrophil/lymphocyte ratios were significantly higher in those in the MIS-C group versus the COVID-19 cohort. How were the results of chest CT or Urine examination indicating bacterial infection, among the 3 groups (MIS-C, COVID-19, and controls)?
*We wonder if the ‘control groups: undergoing routine ambulatory surgery for conditions unrelated to COVID-19 or an inflammatory disorder’ include objects with certain severe infections, and taking immunosuppressive agents such as PSL. Please clarify that.
*Please specify the relation between anti-brain Ab and brain manifestations such as encephalitis.
*The authors used DsDNA, Brain, AQP4, Caspr2, MI, pentraxin, and ANMDAR1, and please explain the reason to measure those. Anti-nuclear antibodies, complement body (C3, C4, CH50), and rheumatoid factors should be measured.
*The authors stored serum of 200 uL. But the assay needs 25 uL. We recommend that the authors keep the serum individually with aliquots of 25 uL.
*Why the IgM and IgA, not IgG antibodies to the SARS-CoV-2 receptor binding domain were significantly higher in the MIS-C group than in the COVID-19 group? Please explain the details with appropriate references.
*page 3, 2.3 assays, line 3, the authors must explain the ‘microsphere immunofluorescence away (20)’ more specifically.
*page 4 last line, before measuring absorbance 450nm, please stop the TMB reaction with acid, and record the time and amount of the reagent.
*Among the IL-4, 5, 6, 8, 10, 13, and 33, IL-8, 10, and 33 do not necessarily represent ‘Th2’. Please explain the details of Th1, Th2, Treg, and other cytokines/chemokines.
*The authors must discuss the mechanism of the difference in the races and severity of COVID-19 (Table 1).
*Table 2, why were IL-15 values different among 3 groups?
* In Figure 1, ‘AutoAb levels with significant differences (Table 9) are shown. MIS-C had the greatest variance within a cohort, and there appears to be a bimodal distribution within the MIS-C cohort.’ This explanation needs more discussion in greater detail.
* Quote references below, and re-write the discussion in association with SARS-CoV-2 infection with systemic autoimmunity/inflammation, which is close to the development of MIS-C.
1. Coronavirus disease-19 triggered systemic lupus erythematous: A novel entity.
Garg Y, Kakria N, Singh S, Jindal AK.
Lung India. 2023 Jan-Feb;40(1):79-81.
2. [COVID-19 as a trigger of autoimmune hepatitis. Case report].
Volchkova EA, Legkova KS, Topchy TB.
Ter Arkh. 2022 Feb 15;94(2):259-264.
3. Risk for uveitis relapse after COVID-19 vaccination.
Zhong Z, Wu Q, Lai Y, Dai L, Gao Y, Liao W, Feng X, Yang P.
J Autoimmun. 2022 Dec;133:102925. doi: 10.1016/j.jaut.2022.102925. Epub 2022 Oct 4.
4. Lazarian G, Quinquenel A, Bellal M, Siavellis J, Jacquy C, Re D, et al. . Autoimmune Haemolytic Anaemia Associated With COVID-19 Infection. Br J Haematol (2020) 190:29–31. 10.1111/bjh.16794 - DOI - PMC – PubMed
5. Coronavirus Disease 2019 and the Thyroid - Progress and Perspectives.
Inaba H, Aizawa T.Front Endocrinol (Lausanne). 2021 Jun 24;12:708333.
Comments on the Quality of English LanguagePlease check the English usage in the manuscript for the revised version of the MS.
Author Response
Reviewer 1:
Top of Form
Comments and Suggestions for Authors
V The authors reported inflammatory and autoimmune manifestations of MIS-C in children.
This study aims to explain why certain SARDS-CoV-2-infected children developed MIS-C.
They speculated that Th2-mediated inflammation can elicit autoantibodies, which may account for some of the differences observed between moderate-severe COVID-19 and MIS-C cohort.
Critiques:
*Curiously, in the Ig isotypes (IgA, IgM, and IgG1-IgG4) directed against three antigenic determinants of SARS-CoV-2 (spike receptor binding domain [RBD], nucleocapsid [NC], and Spike 1), the IgG1 anti-RBD, and some other results were high in the control cohort. Assuming that controls were negative for COVID-19, the results must be explained more clearly.
Controls were negative for acute infection (both COVID and other viral infections) as they were coming in for outpatient surgery. However, they were not excluded if they had a prior history of COVID-19. In addition, COVID vaccines were deployed since December 2020. Therefore, elevated IgG1 anti-RBD antibodies could be explained by previous infection or previous vaccination as explained on pages 2-3.
*In the peripheral blood examination, such as the neutrophil/lymphocyte ratios were significantly higher in those in the MIS-C group versus the COVID-19 cohort. How were the results of chest CT or Urine examination indicating bacterial infection, among the 3 groups (MIS-C, COVID-19, and controls)?
The reviewer raised concerns about other potential causes of high neutrophil/lymphocyte ratios, particularly other sources of bacterial infection. CT scans were rarely performed for clinical indications and no bacterial infections were definitively diagnosed among any of the groups. A comparison of urine test results is not feasible because universal urine testing was not performed by the treating clinicians. Although most MIS-C children had a urinalysis done per clinical protocol, it was rarely done in children with acute COVID-19 and was not done in the control group.
All radiographs and laboratory results were considered by the expert review committee when making cohort group determinations. Thus, if a subject had a co-infection, such as presumed bacterial pneumonia based on an indicative chest CT, a documented bacterial UTI, or bacteremia, that individual would have been excluded from the study as we explain on page 3.
*We wonder if the ‘control groups: undergoing routine ambulatory surgery for conditions unrelated to COVID-19 or an inflammatory disorder’ include objects with certain severe infections, and taking immunosuppressive agents such as PSL. Please clarify that.
Children with known severe infections or inflammatory conditions, whether on immunosuppressive agents or not, were excluded from the control group. The types of surgical procedures were as follows: 38% knee surgery (patellar instability, arthroscopy with lysis of adhesions, lateral meniscus transplant, ACL reconstruction, among others), 19% ophthalmological surgeries (exotropia, esotropia, nystagmus), 12% genital/urological (circumcision, congenital urethral meatal stricture), and ~15% other (ENT: rhinoplasty; general surgery: congenital umbilical hernia repair; GI: endoscopies without documented inflammatory conditions)
*Please specify the relation between anti-brain Ab and brain manifestations such as encephalitis.
On page 13 of the manuscript, we stated “Due to the higher levels of IgG anti-brain Ags and the IgG anti-caspr2, which has been associated with encephalitis (64), we considered some neurologic sequelae associated with MIS-C patients. However, the new definition of MIS-C does not include neurologic symptoms (65). The involvement of autoimmune problems associated with MIS-C remains an issue not yet resolved.”
We did not establish a diagnosis of encephalitis in our patients, but we hypothesized that autoantibodies may contribute to certain neurological manifestations. On page 2 we stated: “Sensory dysfunction in MIS-C is likely caused by neuroinflammation that may lead to disruption of neuronal viability and neural circuits, encephalitis, and/or signaling to cells of peripheral organs (16,17).”
*The authors used DsDNA, Brain, AQP4, Caspr2, MI, pentraxin, and ANMDAR1, and please explain the reason to measure those. Anti-nuclear antibodies, complement body (C3, C4, CH50), and rheumatoid factors should be measured.
We elected not to test anti-nuclear antibodies (ANA) or rheumatoid factor (RF) because they are frequently found to be positive in patients hospitalized with COVID-19 (ANA positive in 30%-50% and RF positive in 20%-60%; doi: 10.1016/j.jtauto.2022.100154). Considering that MIS-C is an extremely rare complication of COVID, we did not believe that these frequently positive assays in acute COVID would provide clinically meaningful information to explain why a tiny proportion progress to MIS-C.
With regard to complement components, previous comprehensive studies have shown that activation markers of the classical, alternative and terminal complement pathways are highly elevated, that activation is largely independent of anti-SARS-CoV-2 humoral immune response, but is strongly associated with markers of macrophage activation (doi: 10.1038/s41598-022-23806-5, doi: 10.1084/jem.20211381). Additional studies on the complement pathway are beyond the scope of this report.
We included analysis of the autoantibodies because of the neuroinflammation that has been reported to be associated with COVID especially those with MIS-C, and because we had experience with these biomarkers because we researched autism. The ant-caspr2 antibody levels in MIS-C cohort were especially interesting in that as referenced (64) it has been associated with autoimmune encephalitis.
*The authors stored serum of 200 uL. But the assay needs 25 uL. We recommend that the authors keep the serum individually with aliquots of 25 uL.
We thank the Reviewer for the recommendation.
*Why the IgM and IgA, not IgG antibodies to the SARS-CoV-2 receptor binding domain were significantly higher in the MIS-C group than in the COVID-19 group? Please explain the details with appropriate references.
As we responded to a previous question, it may have been due to some individuals who have been vaccinated. The assay was clinically approved for use as stated in the manuscript; however, some samples may have spurious cross-reactivity or some sort of false positive due to an unrelated inflammatory response. We have no good explanation for this, nor do we have references to suppose the results. We can only suggest this would be the first and unique finding in this paper. It may relate to recency with which this cohort was seen relative to COVID patients (i.e., this group happened to be seen (samples collected) earlier than the Covid-19 cohort and it's a timing issue (IgM and IgA come up faster), or it may be that, since samples were collected over a relatively broad period, that CoV-2 variants impacted (mutations causing a weaker) IgG response but the broader reactivity of the acute isotypes were less affected by the variant mutations, or it could be due to something inherent (cytokine differences?) in MIS-C patients.
*page 3, 2.3 assays, line 3, the authors must explain the ‘microsphere immunofluorescence away (20)’ more specifically.
We dedicate an entire paragraph in section 2.3 of the Methods section to describe the assay in specific detail. We did add more information about this assay.
*page 4 last line, before measuring absorbance 450nm, please stop the TMB reaction with acid, and record the time and amount of the reagent.
When performing the assays, we followed the manufacturer’s instructions precisely and verified that the results were accurate.
*Among the IL-4, 5, 6, 8, 10, 13, and 33, IL-8, 10, and 33 do not necessarily represent ‘Th2’. Please explain the details of Th1, Th2, Treg, and other cytokines/chemokines.
IL-8, IL-10, and IL-33 are not classically considered type-2 cytokines; however, they have stronger associations with Th2 than Th1 cells. ILC2 and Th2 are associated with eosinophil driven type-2 inflammation. There was eosinophilia in these patients which also has been associated with elevated IL-33 and IL-8 in asthma and allergies (Fahy, J.V. Nat Rev Immunol 2015;15,57-65). The relationship between IL-10 and Th1 and Th2 subsets is indirect: IL-10 selectively inhibits the interaction between antigen presenting cells (APC) and Th1 cells greater than with Th2 cells (Rasquinha M.T. et al, J. Immunol. 2021; 207,2205-2215)
*The authors must discuss the mechanism of the difference in the races and severity of COVID-19 (Table 1).
The control group in our study was 70% White, while the two hospitalized cohorts had significantly fewer Whites, with only 15% of MIS-C and 41% of COVID+ patients. This discrepancy reflects the patient demographics of our inpatient and outpatient clinics in Connecticut. Our outpatient and surgical subspecialties draw patients from across the state, whereas our inpatient unit primarily serves Hartford County, which has a more diverse racial/ethnic makeup. Similar findings were published in 2024, where an outpatient mild-COVID cohort was predominantly White (90).
Multiple potential explanations exist for the differences observed in the hospitalized cohorts (COVID and MIS-C). Both conditions have known racial/ethnic disparities in their distribution. According to Vicetti Miguel et al. (91), racial/ethnic differences in the incidence of COVID-19 and MIS-C are well-documented. Encinosa et al. (92) found that MIS-C incidence varies twofold by race, with 0.97 per 100,000 in White children compared to 1.99 per 100,000 in non-Hispanic Black children. Similarly, COVID-19 incidence varied from 4.4 per 100,000 in White children to 6.6 per 100,000 in Black children. This has been added to the limitations of our study on page 14-15.
It must be noted that the children enrolled in our study do not necessarily reflect the makeup of all admitted children with MIS-C and COVID, but rather a selected population of enrollees. We had a near-universal approach to children suspected of having MIS-C due to its rarity. However, given the overwhelming number of children admitted with acute COVID at times, not all were approached for enrollment. Additionally, well-described differences in racial/ethnic clinical trial enrollments in pediatric research may also explain discrepancies in our enrollments.
*Table 2, why were IL-15 values different among 3 groups?
IL-15 was significantly higher in the sera of COVID+ and MIS-C patients than in the Controls. Although the mean level was higher with MIS-C than COVID+, it was not significant likely due to the small number of subjects. These findings recapitulate those of other studies and highlight the characteristic and unique inflammatory host response in children with MIS-C.
* In Figure 1, “AutoAb levels with significant differences (Table 9) are shown. MIS-C had the greatest variance within a cohort, and there appears to be a bimodal distribution within the MIS-C cohort”. This explanation needs more discussion in greater detail.
We investigated possible reasons for the bimodal distribution of antibody levels to brain antigens, caspr2 and pentraxin by comparing the distributions of age, sex and year of enrollment in patients that clustered the higher vs lower antibody ranges. However, we could not find significant differences. This observation will require a larger cohort in future studies to investigate the differences in greater detail as we stated on page 8.
* Quote references below, and re-write the discussion in association with SARS-CoV-2 infection with systemic autoimmunity/inflammation, which is close to the development of MIS-C.
The request for inclusion of these references which relate to autoimmunity suggests to us that you are accepting our suggestion that type-2 inflammation and autoantibodies are influencing the immunity in the MIS-C cohort more than that of the COVID+ cohort. This relates to the question about certain antibody analyses that we performed e.g., the anti-caspr2. Please see our response to your question. Our discussion has already emphasized the COVID-19 and MIS-C connections with autoimmunity. Although we had already also included several references relating to the association, we have now included the suggested references and added more references and discussion of the connection of some autoimmune aspects of MIS-C.
Reviewer 2 Report
Comments and Suggestions for Authors
I thank the author for a logical and interesting study, which describes not only the groups themselves, but also the train of thought during the study, as well as adjustments to the study according to the results obtained. This adds meaning and improves performance.
Recommendations to authors:
1. Illustrate with drawings the possible mechanisms of inflammation that are characteristic of MIS-C.
2. Place the tables along the text, so they will be easier to read.
3. Discuss the role of genetic variants in the pathogenesis of MIS-C.
Questions:
1. Why did the authors choose only moderate and severe forms of COVID19 for comparison and excluded mild and moderate ones? Are mild to moderate cases not hospitalized in the USA?
2. How new is the data obtained (IL level), how do they fundamentally differ from those that were performed previously?
3. Samples were collected from April 1, 2020 to July 1, 2022. At this time there were different strains of the virus. Has strain dependence been analyzed?
4. Were repeated cases of infection taken into account? I would like to see the inclusion and exclusion criteria in more detail.
5. How was COVID19 infection excluded or even excluded in the control group?
6. I would like some clarification about asthma. Why was it observed so often in all patients?
Author Response
Reviewer 1:
Top of Form
Comments and Suggestions for Authors
V The authors reported inflammatory and autoimmune manifestations of MIS-C in children.
This study aims to explain why certain SARDS-CoV-2-infected children developed MIS-C.
They speculated that Th2-mediated inflammation can elicit autoantibodies, which may account for some of the differences observed between moderate-severe COVID-19 and MIS-C cohort.
Critiques:
*Curiously, in the Ig isotypes (IgA, IgM, and IgG1-IgG4) directed against three antigenic determinants of SARS-CoV-2 (spike receptor binding domain [RBD], nucleocapsid [NC], and Spike 1), the IgG1 anti-RBD, and some other results were high in the control cohort. Assuming that controls were negative for COVID-19, the results must be explained more clearly.
Controls were negative for acute infection (both COVID and other viral infections) as they were coming in for outpatient surgery. However, they were not excluded if they had a prior history of COVID-19. In some cases, it may have been very mild and not reported to a physician. In addition, COVID-19 vaccines have been deployed since December 2020. Therefore, elevated IgG anti-RBD antibodies could be explained by previous infection or previous vaccination as explained on pages 2-3.
*In the peripheral blood examination, such as the neutrophil/lymphocyte ratios were significantly higher in those in the MIS-C group versus the COVID-19 cohort. How were the results of chest CT or Urine examination indicating bacterial infection, among the 3 groups (MIS-C, COVID-19, and controls)?
The reviewer raised concerns about other potential causes of high neutrophil/lymphocyte ratios, particularly other sources of bacterial infection. CT scans were rarely performed for clinical indications and no bacterial infections were definitively diagnosed among any of the groups. A comparison of urine test results is not feasible because universal urine testing was not performed by the treating clinicians. Although most MIS-C children had a urinalysis done per clinical protocol, it was rarely done in children with acute COVID-19 and was not done in the control group.
All radiographs and laboratory results were considered by the expert review committee when making cohort group determinations. Thus, if a subject had a co-infection, such as presumed bacterial pneumonia based on an indicative chest CT, a documented bacterial UTI, or bacteremia, that individual would have been excluded from the study as we explain on page 3.
*We wonder if the ‘control groups: undergoing routine ambulatory surgery for conditions unrelated to COVID-19 or an inflammatory disorder’ include objects with certain severe infections, and taking immunosuppressive agents such as PSL. Please clarify that.
Children with known severe infections or inflammatory conditions, whether on immunosuppressive agents or not, were excluded from the control group. The types of surgical procedures were as follows: 38% knee surgery (patellar instability, arthroscopy with lysis of adhesions, lateral meniscus transplant, ACL reconstruction, among others), 19% ophthalmological surgeries (exotropia, esotropia, nystagmus), 12% genital/urological (circumcision, congenital urethral meatal stricture), and ~15% other (ENT: rhinoplasty; general surgery: congenital umbilical hernia repair; GI: endoscopies without documented inflammatory conditions)
*Please specify the relation between anti-brain Ab and brain manifestations such as encephalitis.
On page 13 of the manuscript, we stated “Due to the higher levels of IgG anti-brain Ags and the IgG anti-caspr2, which has been associated with encephalitis (64), we considered some neurologic sequelae associated with MIS-C patients. However, the new definition of MIS-C does not include neurologic symptoms (65). The involvement of autoimmune problems associated with MIS-C remains an issue not yet resolved.”
We did not establish a diagnosis of encephalitis in our patients, but we hypothesized that autoantibodies may contribute to certain neurological manifestations. On page 2 we stated: “Sensory dysfunction in MIS-C is likely caused by neuroinflammation that may lead to disruption of neuronal viability and neural circuits, encephalitis, and/or signaling to cells of peripheral organs (16,17).”
*The authors used DsDNA, Brain, AQP4, Caspr2, MI, pentraxin, and ANMDAR1, and please explain the reason to measure those. Anti-nuclear antibodies, complement body (C3, C4, CH50), and rheumatoid factors should be measured.
We elected not to test anti-nuclear antibodies (ANA) or rheumatoid factor (RF) because they are frequently found to be positive in patients hospitalized with COVID-19 (ANA positive in 30%-50% and RF positive in 20%-60%; doi: 10.1016/j.jtauto.2022.100154). Considering that MIS-C is an extremely rare complication of COVID, we did not believe that these frequently positive assays in acute COVID would provide clinically meaningful information to explain why a tiny proportion progress to MIS-C.
With regard to complement components, previous comprehensive studies have shown that activation markers of the classical, alternative and terminal complement pathways are highly elevated, that activation is largely independent of anti-SARS-CoV-2 humoral immune response, but is strongly associated with markers of macrophage activation (doi: 10.1038/s41598-022-23806-5, doi: 10.1084/jem.20211381). Additional studies on the complement pathway are beyond the scope of this report.
We included analysis of the autoantibodies to caspr2, AQP4, NMDAR1, etc, because of the neuroinflammation that has been reported to be associated with COVID especially those with MIS-C, and because we had experience with these biomarkers because we researched autism. Anti-caspr2 antibodies as referenced (64) has been useful in diagnosing some patients with autoimmune encephalitis. Our results were very interesting in that the MIS-cohort had a significant difference from both the COVID+ and Control cohorts.
*The authors stored serum of 200 uL. But the assay needs 25 uL. We recommend that the authors keep the serum individually with aliquots of 25 uL.
We thank the Reviewer for the recommendation.
*Why the IgM and IgA, not IgG antibodies to the SARS-CoV-2 receptor binding domain were significantly higher in the MIS-C group than in the COVID-19 group? Please explain the details with appropriate references.
As we responded to a previous question, it may have been due to some individuals who had been vaccinated. The assay was clinically approved for use as stated in the manuscript; however, some samples may have spurious cross-reactivity or some sort of false positivity due to an unrelated inflammatory response. We have no good explanation for this, nor do we have references to suppose the results. We can only suggest this would be the first and unique finding in this paper. It may relate to recency with which this cohort was seen relative to COVID patients (i.e., this group happened to be seen (samples collected) earlier than the Covid-19 cohort and it's a timing issue (IgM and IgA come up faster), or it may be that, since samples were collected over a relatively broad period, that CoV-2 variants impacted (mutations causing a weaker) IgG response but the broader reactivity of the acute isotypes were less affected by the variant mutations, or it could be due to something inherent (cytokine differences?) in MIS-C patients.
*page 3, 2.3 assays, line 3, the authors must explain the ‘microsphere immunofluorescence away (20)’ more specifically.
We dedicate an entire paragraph in section 2.3 of the Methods section to describe the assay in specific detail. We did add more information about this assay on page 3.
*page 4 last line, before measuring absorbance 450nm, please stop the TMB reaction with acid, and record the time and amount of the reagent.
When performing the assays, we followed the manufacturer’s instructions precisely and verified that the results were accurate.
*Among the IL-4, 5, 6, 8, 10, 13, and 33, IL-8, 10, and 33 do not necessarily represent ‘Th2’. Please explain the details of Th1, Th2, Treg, and other cytokines/chemokines.
IL-8, IL-10, and IL-33 are not classically considered type-2 cytokines; however, they have stronger associations with Th2 than Th1 cells. ILC2 and Th2 are associated with eosinophil driven type-2 inflammation. There was eosinophilia in these patients which also has been associated with elevated IL-33 and IL-8 in asthma and allergies (Fahy, J.V. Nat Rev Immunol 2015;15,57-65). The relationship between IL-10 and Th1 and Th2 subsets is indirect: IL-10 selectively inhibits the interaction between antigen presenting cells (APC) and Th1 cells greater than with Th2 cells (Rasquinha M.T. et al, J. Immunol. 2021; 207,2205-2215)
*The authors must discuss the mechanism of the difference in the races and severity of COVID-19 (Table 1).
The control group in our study was 70% White, while the two hospitalized cohorts had significantly fewer Whites, with only 15% of MIS-C and 41% of COVID+ patients. This discrepancy reflects the patient demographics of our inpatient and outpatient clinics in Connecticut. Our outpatient and surgical subspecialties draw patients from across the state, whereas our inpatient unit primarily serves Hartford County, which has a more diverse racial/ethnic makeup. Similar findings were published in 2024, where an outpatient mild-COVID cohort was predominantly White (90).
Multiple potential explanations exist for the differences observed in the hospitalized cohorts (COVID and MIS-C). Both conditions have known racial/ethnic disparities in their distribution. According to Vicetti Miguel et al. (91), racial/ethnic differences in the incidence of COVID-19 and MIS-C are well-documented. Encinosa et al. (92) found that MIS-C incidence varies twofold by race, with 0.97 per 100,000 in White children compared to 1.99 per 100,000 in non-Hispanic Black children. Similarly, COVID-19 incidence varied from 4.4 per 100,000 in White children to 6.6 per 100,000 in Black children. This has been added to the limitations of our study on page 14-15.
It must be noted that the children enrolled in our study do not necessarily reflect the makeup of all admitted children with MIS-C and COVID, but rather a selected population of enrollees. We had a near-universal approach to children suspected of having MIS-C due to its rarity. However, given the overwhelming number of children admitted with acute COVID at times, not all were approached for enrollment. Additionally, well-described differences in racial/ethnic clinical trial enrollments in pediatric research may also explain discrepancies in our enrollments.
*Table 2, why were IL-15 values different among 3 groups?
IL-15 was significantly higher in the sera of COVID+ and MIS-C patients than in the Controls. Although the mean level was higher with MIS-C than COVID+, it was not significant, likely due to the small number of subjects. These findings recapitulate those of other studies and highlight the characteristic and unique inflammatory host response in children with MIS-C.
* In Figure 1, “AutoAb levels with significant differences (Table 9) are shown. MIS-C had the greatest variance within a cohort, and there appears to be a bimodal distribution within the MIS-C cohort”. This explanation needs more discussion in greater detail.
We investigated possible reasons for the bimodal distribution of the autoantibody levels to brain antigens, caspr2 and pentraxin by comparing the distributions of age, sex and year of enrollment in patients that clustered the higher vs lower antibody ranges. However, we could not find significant differences. This observation will require a larger cohort in future studies to investigate the differences in greater detail as we stated on page 8.
* Quote references below, and re-write the discussion in association with SARS-CoV-2 infection with systemic autoimmunity/inflammation, which is close to the development of MIS-C.
The request for inclusion of these references which relate to autoimmunity suggests to us that you are accepting our suggestion that type-2 inflammation and autoantibodies are influencing the immunity in the MIS-C cohort more than that of the COVID+ cohort. This relates to the question about certain antibody analyses that we performed e.g., the anti-caspr2. Please see our response to your question. Our discussion has already emphasized the COVID-19 and MIS-C connections with autoimmunity. Although we had already also included several references relating to the association, we have now included the suggested references and added more references and discussion of the connection of some autoimmune aspects of MIS-C.
Round 2
Reviewer 1 Report
Comments and Suggestions for Authors
The manuscript was appropriately improved.